# Diagnostic Performance and Prognostic Value of PET/CT with Different Tracers for Brain Tumors: A Systematic Review of Published Meta-Analyses

**DOI:** 10.3390/ijms20194669

**Published:** 2019-09-20

**Authors:** Giorgio Treglia, Barbara Muoio, Gianluca Trevisi, Maria Vittoria Mattoli, Domenico Albano, Francesco Bertagna, Luca Giovanella

**Affiliations:** 1Clinic of Nuclear Medicine and PET/CT Center, Imaging Institute of Southern Switzerland, Ente Ospedaliero Cantonale, CH-6500 Bellinzona, Switzerland; luca.giovanella@eoc.ch; 2Health Technology Assessment Unit, Academic Education, Research and Innovation Area, General Directorate, Ente Ospedaliero Cantonale, CH-6500 Bellinzona, Switzerland; 3Department of Nuclear Medicine and Molecular Imaging, Lausanne University Hospital and University of Lausanne, CH-1011 Lausanne, Switzerland; 4Clinic of Medical Oncology, Oncology Institute of Southern Switzerland, Ente Ospedaliero Cantonale, CH-6500 Bellinzona, Switzerland; barbara.muoio@eoc.ch; 5Neurosurgical Unit, Presidio Ospedaliero Santo Spirito, IT-65124 Pescara, Italy; trevisi.gianluca@gmail.com; 6Department of Neurosciences, Imaging and Clinical Sciences, “G. D’Annunzio” University, IT-66100 Chieti, Italy; mvittoriamattoli@yahoo.it; 7Department of Nuclear Medicine, Spedali Civili of Brescia and University of Brescia, IT-25123 Brescia, Italy; doalba87@libero.it (D.A.); francesco.bertagna@unibs.it (F.B.); 8Department of Nuclear Medicine, University Hospital Zurich and University of Zurich, CH-8091 Zurich, Switzerland

**Keywords:** PET, positron emission tomography, brain tumors, glioma, brain metastases, diagnostic performance, prognosis, meta-analysis

## Abstract

Background: Several meta-analyses reporting data on the diagnostic performance or prognostic value of positron emission tomography (PET) with different tracers in detecting brain tumors have been published so far. This review article was written to summarize the evidence-based data in these settings. Methods: We have performed a comprehensive literature search of meta-analyses published in the Cochrane library and PubMed/Medline databases (from inception through July 2019) about the diagnostic performance or prognostic value of PET with different tracers in patients with brain tumors. Results: We have summarized the results of 24 retrieved meta-analyses on the use of PET or PET/computed tomography (CT) with different tracers in brain tumors. The tracers included were: fluorine-18 fluorodeoxyglucose (^18^F-FDG), carbon-11 methionine (^11^C-methionine), fluorine-18 fluoroethyltyrosine (^18^F-FET), fluorine-18 dihydroxyphenylalanine (^18^F-FDOPA), fluorine-18 fluorothymidine (^18^F-FLT), and carbon-11 choline (^11^C-choline). Evidence-based data demonstrated good diagnostic performance of PET with different tracers in detecting brain tumors, in particular, radiolabelled amino acid tracers showed the highest diagnostic performance values. All the PET tracers evaluated had significant prognostic value in patients with glioma. Conclusions: Evidence-based data showed a good diagnostic performance for some PET tracers in specific indications and significant prognostic value in brain tumors.

## 1. Introduction

Positron emission tomography (PET) is a nuclear medicine imaging technique that, using different radiotracers evaluating different metabolic patterns, is able to detect in advance pathophysiological changes in oncological patients, including those with brain tumors. These functional changes usually occur before the development of morphological changes detected by conventional radiological imaging techniques such as computed tomography (CT) and magnetic resonance imaging (MRI) [1]. MRI is the standard neuroimaging method used for diagnosis of brain tumors, for performing stereotactic biopsy, and for surgical planning in neuro-oncology [2]. Currently, hybrid imaging techniques such as PET/CT and PET/MRI, providing a combination of both functional and morphological information, may be useful methods for early diagnosis of brain tumors [1,2].

Different PET radiotracers have been used to evaluate brain tumors, including fluorine-18 fluorodeoxyglucose (^18^F-FDG), carbon-11 methionine (^11^C-methionine), fluorine-18 fluoroethyltyrosine (^18^F-FET), fluorine-18 fluorodihydroxyphenylalanine (^18^F-FDOPA), fluorine-18 fluorothymidine (^18^F-FLT), and radiolabelled choline (^11^C-choline or ^18^F-choline).

^18^F-FDG is the most used PET radiotracer in oncology; it is a radiolabelled glucose analogue taken up by neoplastic cells via cell membrane glucose transporters (GLUT) and subsequently phosphorylated through the activity of intracellular hexokinase. ^18^F-FDG allows the detection of neoplastic cells due to their frequently increased glucose metabolism [1]. The main concerns about the use of ^18^F-FDG PET for evaluating brain lesions are the high background ^18^F-FDG uptake in the normal brain [1] and the increased ^18^F-FDG uptake in inflammatory or benign lesions [3]. Therefore, other PET tracers characterized by increased uptake in brain tumors and low uptake in the normal brain have been developed in recent decades [1].

^11^C-methionine is a radiolabelled amino acid; methionine is used by the cells in two main metabolic functions: protein synthesis and conversion to *S*-adenosylmethionine. In many neoplastic cells, there is an increase in protein synthesis, transmethylation and transsulfuration, leading to an increased uptake of ^11^C-methionine [1,4]. Unfortunately, the use of this PET tracer is restricted to PET centers with a cyclotron facility because of the shorter half-life of ^11^C compared to ^18^F (20 min versus 110 min, respectively) [4].

^18^F-FET is a fluorinated amino acid used to detect brain tumors. ^18^F-FET is taken up into neoplastic cells due to their increased amino acid uptake through an L-type amino acid transport system, and it is not incorporated into proteins [2].

^18^F-FDOPA has been proposed as a useful PET tracer for imaging brain tumors. ^18^F-FDOPA is transported across the blood-brain barrier by a number of amino acid transporters, which have been shown to be overexpressed in brain tumors. After intracellular uptake through the large amino acid transporter, ^18^F-FDOPA is decarboxylated by DOPA decarboxylase to ^18^F-dopamine, which is transported into storage granules by vesicular monoamine transporters and trapped intracellularly [1,5].

^18^F-FLT is a tracer proposed as an imaging biomarker of cell proliferation, which is increased in neoplastic cells; during the S phase of the cell cycle, ^18^F-FLT is phosphorylated by thymidine-kinase-1 and trapped inside the cell but not incorporated into the DNA. The cellular thymidine-kinase-1 activity has been reported to be proportional to the proliferation activity of the tumor [6].

Lastly, as tumor cells present a high turnover of cellular membranes, radiolabelled choline (using ^11^C or ^18^F) may be used to detect brain tumors. The uptake of radiolabelled choline increases in tumor tissue to keep up with the demands of phospholipids synthesis in cellular membranes [1,7].

Enough literature data already exist about the diagnostic performance and prognostic value of PET with different tracers in brain tumors. Therefore, we aimed to summarize the findings of published meta-analyses in these settings.

## 2. Methods

A comprehensive computer literature search of the Cochrane library and PubMed/Medline databases was performed to find published meta-analyses on the diagnostic performance or prognostic value of PET or hybrid PET/CT or PET/MRI in patients with brain tumors. The search algorithm is reported in the Appendix A. The literature search was updated until July 31st, 2019. No language restriction was used. Meta-analyses investigating the diagnostic performance or prognostic value of PET or PET/CT or PET/MRI with different tracers in brain tumors were eligible for inclusion. We reviewed titles and abstracts of the retrieved articles, applying the selected inclusion criteria. For each selected meta-analysis, we collected information about basic study characteristics (authors, year of publication, number of original studies included, number of patients included, clinical indications, and radiotracer used) and pooled diagnostic performance measures (sensitivity, specificity, positive and negative likelihood ratios, and diagnostic odds ratios) including 95% confidence interval values (95% CI). Moreover, we have briefly described the main findings of the selected diagnostic or prognostic meta-analyses in the results section.

## 3. Results

Twenty-four meta-analyses on the use of PET or PET/CT with different tracers in brain tumors, published from 2012, were selected through the comprehensive computer literature search (Figure 1) [8,9,10,11,12,13,14,15,16,17,18,19,20,21,22,23,24,25,26,27,28,29,30,31]. The characteristics of the selected meta-analyses on the diagnostic performance are presented in Table 1. Here, below, we have summarized the main findings of the meta-analytic studies based on the different clinical indications of PET or PET/CT.

### 3.1. Evaluation of Suspicious Primary Brain Tumor

Four meta-analyses have assessed the diagnostic performance of PET or PET/CT with different tracers in patients for whom primary brain tumors are suspected [8,25,28,31]. Pooled results are showed in Figure 2.

#### 3.1.1. ^18^F-FDG

A meta-analysis including patients with suspicious primary brain tumors showed that ^18^F-FDG PET or PET/CT has a moderate sensitivity and specificity for differentiating brain tumors from non-tumor lesions. False-positive findings were often due to inflammatory lesions or other non-tumor tissues; on the other hand, reduced ^18^F-FDG uptake in brain tumors is usually influenced by the high physiological glucose metabolism in the surrounding normal brain tissue, leading to a decreased sensitivity [28]. Another meta-analysis also demonstrated that ^18^F-FDG PET or PET/CT have a moderate diagnostic performance in distinguishing between tumoral and non-tumoral lesions in the brain, lower than amino acid PET [25].

#### 3.1.2. ^11^C-Methionine

A meta-analysis by Zhao et al. demonstrated a good diagnostic performance of ^11^C-methionine PET or PET/CT in detecting brain tumors (pooled sensitivity and specificity were of 95% and 83%, respectively) with higher diagnostic accuracy values compared to ^18^F-FDG PET or PET/CT, likely due to the higher ^11^C-methionine uptake in brain tumors and lower accumulation in normal brain tissue [28].

#### 3.1.3. ^18^F-FET

For initial assessment of patients with a newly diagnosed brain lesion, ^18^F-FET PET or PET/CT demonstrated a good performance in the diagnosis of a brain tumor with a pooled sensitivity and specificity of 82% and 76%, respectively. A mean tumor-to-background uptake ratio (TBR) threshold of at least 1.6 and a maximum TBR of at least 2.1 had the best diagnostic value for differentiating brain tumors from non-tumoral brain lesions. For the diagnosis of glioma versus non-glioma brain lesions, ^18^F-FET PET or PET/CT demonstrated a good sensitivity (84%) but an inadequate specificity (62%) [31]. In a head-to-head comparative meta-analysis, the diagnostic performance of ^18^F-FET PET or PET/CT in distinguishing between tumoral and non-tumoral lesions in the brain was found to be significantly higher compared to that of ^18^F-FDG PET or PET/CT performed in the same patients [25].

#### 3.1.4. ^18^F-FDOPA

^18^F-FDOPA PET or PET/CT revealed a moderate pooled sensitivity (71%) and a good pooled specificity (86%) in detecting newly-diagnosed gliomas [8].

### 3.2. Glioma Grading

Gliomas are the most frequently occurring primary brain tumors. High grade gliomas like glioblastomas are the most common gliomas in adults, with a poor prognosis with any current therapy. Conversely, low-grade gliomas, the second most common type of gliomas, are potentially curable with appropriate treatment. Several meta-analyses have evaluated the role of PET or PET/CT with different tracers in differentiating between high-grade and low-grade gliomas [8,11,19,25]. Pooled results are showed in Figure 3.

#### 3.2.1. ^18^F-FDG

^18^F-FDG uptake is significantly higher in high-grade gliomas compared to low-grade gliomas. According to the meta-analysis of Dunet et al., a mean TBR of at least 1.4 and a maximum TBR of at least 1.8 at ^18^F-FDG PET had the best value to distinguish between low- and high-grade gliomas, with a sensitivity, specificity and accuracy of 60%, 91%, and 74%, respectively, for mean TBR and 72%, 73%, and 72%, respectively, for maximum TBR [25]. A recent meta-analysis demonstrated a lower sensitivity of ^18^F-FDG PET or PET/CT in differentiating between high-grade and low-grade gliomas compared to radiolabelled amino acid PET (^11^C-methionine and ^18^F-FET) but with higher specificity [11].

#### 3.2.2. ^11^C-Methionine

^11^C-methionine PET or PET/CT had a moderate diagnostic accuracy in differentiating between high-grade and low-grade gliomas, according to data provided by a recent meta-analysis (pooled sensitivity and specificity of 80% and 72%, respectively) [19]. Another meta-analysis demonstrated that ^11^C-methionine PET or PET/CT has a higher sensitivity compared to ^18^F-FDG PET or PET/CT in differentiating between high-grade and low-grade gliomas but with lower specificity; diagnostic performance values were similar to those of ^18^F-FET PET or PET/CT in this setting [11].

#### 3.2.3. ^18^F-FET

^18^F-FET uptake is significantly higher in high-grade gliomas compared with low-grade gliomas. Dunet et al. reported that a mean TBR of at least 2.0 and a maximum TBR of at least 3.0 at ^18^F-FET PET reached a sensitivity, specificity, and accuracy of 88%, 73%, and 81%, respectively, for mean TBR, and 80%, 82%, and 81%, respectively, for maximum TBR [25]. A recent meta-analysis demonstrated that ^18^F-FET PET or PET/CT has a higher sensitivity compared to ^18^F-FDG PET or PET/CT in differentiating between high-grade and low-grade gliomas but with lower specificity; diagnostic performance values were similar to those of ^11^C-methionine PET or PET/CT in this setting [11].

#### 3.2.4. ^18^F-FDOPA

For differentiating high-grade from low-grade gliomas, ^18^F-FDOPA PET or PET/CT showed a pooled sensitivity of 88% and a pooled specificity of 73% [8].

### 3.3. Delineation of Gliomas

For surgical and radiation therapy planning in patients with glioma, a correct delineation of the target volume is needed. A recent evidence-based article suggested that radiolabelled amino acid PET may ameliorate the delineation of high-grade gliomas compared to standard MRI [21].

### 3.4. Diagnosis of Recurrent Brain Tumors

Distinguishing recurrent brain tumors from non-tumoral lesions after radiation therapy and/or chemotherapy is a crucial clinical issue, because the different diagnosis will lead to divergent treatments. Several meta-analyses have assessed the diagnostic performance of PET with different tracers in this setting [8,9,12,14,16,18,26,27,28,29,30]. Pooled results are showed in Figure 4.

#### 3.4.1. ^18^F-FDG

A meta-analysis by Zhao et al. demonstrated a moderate diagnostic accuracy of ^18^F-FDG PET or PET/CT in detecting brain tumor recurrence [28]. This finding was confirmed by another meta-analysis, which showed a pooled sensitivity and specificity of 78% and 77%, respectively [26]. Furuse et al. showed that the diagnostic performance of ^18^F-FDG PET or PET/CT in detecting recurrent brain tumors was lower compared to that of radiolabelled amino acid PET or PET/CT [12]. Nihashi et al. showed that, when considering both low- and high-grade gliomas, pooled sensitivity and specificity of ^18^F-FDG PET or PET/CT in detecting glioma recurrence were 77% and 78%, respectively. In subgroup analyses limited to high-grade gliomas, pooled sensitivity and specificity were 79% and 70%, respectively [30]. Wang et al. reported a moderate sensitivity (70%) but a good specificity (88%) of ^18^F-FDG PET or PET/CT in detecting recurrent glioma; however, the diagnostic accuracy was lower compared to that of ^11^C-methionine PET or PET/CT and magnetic resonance spectroscopy in this setting [27]. Another meta-analysis demonstrated that the diagnostic performance of ^18^F-FDG PET or PET/CT in detecting recurrent glioma is not optimal, in particular if compared with other available neuroimaging methods [12].

#### 3.4.2. ^11^C-Methionine

^11^C-methionine PET or PET/CT demonstrated good diagnostic performance in detecting brain tumor recurrence (pooled sensitivity and specificity of 92% and 87%, respectively), with higher values compared to ^18^F-FDG PET or PET/CT [28]. For high-grade gliomas, pooled sensitivity and specificity of ^11^C-methionine PET or PET/CT in detecting glioma recurrence were 70% and 93%, respectively [30]. Compared to dynamic susceptibility contrast-enhanced MRI, ^11^C-methionine PET or PET/CT demonstrated comparable pooled sensitivity and specificity in detecting glioma recurrence, with pooled values of 87% and 81.3%, respectively [29]. Similar values of sensitivity and specificity (85% and 83%, respectively) were described by Wang et al. who demonstrated that the diagnostic performance of ^11^C-methionine PET or PET/CT in detecting glioma recurrence was similar to that of magnetic resonance spectroscopy [27]. A large meta-analysis including 29 studies confirmed the good diagnostic performance of ^11^C-methionine PET or PET/CT in this setting, with a pooled sensitivity and specificity of 88% and 85%, respectively [18].

#### 3.4.3. ^18^F-FET

A recent meta-analysis demonstrated that ^18^F-FET PET or PET/CT has a good diagnostic accuracy in differentiating between brain tumor recurrence from radiation necrosis after treatment, with pooled sensitivity and specificity values of 82% and 80%, respectively. In the subgroup of patients with suspicious glioma recurrence, sensitivity and specificity of ^18^F-FET PET or PET/CT were 83% and 81%, respectively [16]. The good diagnostic performance of ^18^F-FET PET or PET/CT in this setting was also confirmed by Furuse et al., who reported increased diagnostic performance of ^18^F-FET PET or PET/CT compared to ^18^F-FDG and ^11^C-methionine PET or PET/CT [12]. Kim et al. found that amino acid PET or PET/CT, including ^18^F-FET PET, has a good diagnostic performance in differentiating residual or recurrent brain tumors from treatment-related changes (pseudoprogression) in patients with high-grade gliomas [9].

#### 3.4.4. ^18^F-FDOPA

A recent meta-analysis indicated that ^18^F-FDOPA PET or PET/CT has a good diagnostic accuracy in differentiating between brain tumor recurrence from radiation necrosis after treatment, with pooled sensitivity and specificity values of 85% and 77%, respectively. In the subgroup of patients with suspicious glioma recurrence, sensitivity and specificity of ^18^F-FDOPA PET or PET/CT were 94% and 89%, respectively [16]. Xiao et al. reported a good sensitivity of ^18^F-FDOPA PET and PET/CT in detecting recurrent glioma (92%) and a moderate specificity (76%) [8].

#### 3.4.5. ^18^F-FLT

^18^F-FLT PET or PET/CT demonstrated a similar diagnostic performance in detecting brain tumor recurrence compared to ^18^F-FDG PET or PET/CT, with pooled sensitivity and specificity of 82% and 76%, respectively [26].

#### 3.4.6. ^18^C-Choline

A recent meta-analysis indicated that ^11^C-choline PET or PET/CT has a good diagnostic accuracy for differentiating glioma recurrence from radiation induced necrosis after treatment, with a pooled sensitivity and specificity of 87% and 82%, respectively [14].

### 3.5. Diagnosis of Brain Metastases

The reliability of PET or PET/CT with different tracers in detecting brain metastases has been evaluated to a less extent compared to primary brain tumors. A meta-analysis demonstrated that the pooled sensitivity and specificity of ^18^F-FDG PET or PET/CT in detecting brain metastases in patients with lung cancer were 21% and 100%, respectively. In particular, the sensitivity of this method is lower compared to that of contrast-enhanced MRI [24].

### 3.6. Diagnosis of Recurrent Brain Metastases

The meta-analysis by Li et al., which focused on the use of PET or PET/CT with different tracers in differentiating recurrent brain metastasis from radionecrosis after radiation therapy, demonstrated a good diagnostic accuracy of PET or PET/CT with both ^18^F-FDG and radiolabelled amino acid tracers (^11^C-methionine, ^18^F-FET, ^18^F-FDOPA) in this setting [17]. MRI and PET with different tracers showed similar diagnostic performance for the detection of recurrent brain metastasis after stereotactic radiosurgery; nevertheless, advanced MRI methods showed a significantly higher diagnostic performance in this setting compared to PET [13]. Pooled results are showed in Figure 5.

### 3.7. Diagnosis of Primary Central Nervous System Lymphoma (PCNSL)

^18^F-FDG PET and PET/CT showed considerable accuracy in identifying PCNSL among various brain lesions in immunocompetent patients (pooled sensitivity and specificity of 88% and 86%, respectively), therefore, ^18^F-FDG PET/CT could be a valuable diagnostic imaging method in this setting [22]. High diagnostic accuracy of ^18^F-FDG PET and PET/CT has also been demonstrated in identifying PCNSL among various brain lesions in patients with human immunodeficiency virus (HIV) infection [23].

### 3.8. Prognostic Value In Patients With Glioma

Beyond the diagnostic accuracy, PET/CT parameters, and particularly the TBR, may be significant prognostic factors in patients with glioma. A recent meta-analysis demonstrated that increased TBR at ^18^F-FDG PET, ^11^C-methionine PET and ^18^F-FET PET could indicate poor overall survival (pooled hazard ratios were 3.05 for ^18^F-FDG PET, 1.59 for ^11^C-methionine PET, and 1.15 for ^18^F-FET PET) [10]. Another meta-analysis showed that the TBR and metabolic tumor volume at ^11^C-methionine PET are significant prognostic parameters for patients with gliomas. Patients with a high TBR have a higher risk of death, and patients with a high metabolic tumor volume have a higher risk of adverse events or death [15].

## 4. Discussion

Overall, there are increased evidence-based data about the usefulness of PET or PET/CT with different tracers in patients with brain tumors. MRI provides accurate morphological information on brain lesions, whereas PET/CT, using tracers evaluating different metabolic pathways, provides useful information that may definitively improve the diagnostic accuracy of brain tumors if combined with MRI [20,32]. In particular, current evidence based data suggest that radiolabelled amino acid PET or PET/CT is an accurate diagnostic method for several clinical indications including evaluation of suspicious brain tumors, glioma grading and delineation, and detection of brain tumor recurrence. Furthermore, PET/CT with ^18^F-FDG or radiolabelled amino acids may provide useful prognostic information in patients with brain tumors.

In addition to the different characteristics of the various PET tracers, different imaging modalities, such as PET alone compared to PET/CT or PET/MRI, will affect the sensitivity and specificity. In fact, higher diagnostic accuracy values can be obtained by using hybrid modalities compared to PET alone. Overall, the studies in the literature, in various degree, suggested that PET/CT with different tracers may show brain tumor boundaries better than conventional MRI. But compared to PET/CT, conventional MRI can more clearly show the anatomical structure, which is a function that cannot be replaced by PET/CT with any tracer. Therefore, PET/CT and MRI are often combined to evaluate brain tumors in order to achieve a higher accuracy compared to single imaging methods [20].

Awareness of the results described by several published meta-analyses on this topic has the potential to affect patient care by providing supportive evidence-based data for a more effective use of PET/CT with different tracers in the diagnosis of brain tumors. In this regard, recent published international guidelines support nuclear medicine practitioners in recommending, performing, interpreting, and reporting the results of PET/CT with different tracers in patients with glioma [32]. PET/CT with different tracers may be useful to direct therapeutic strategies improving patient outcome in patients with brain tumors; nevertheless, prospective outcome studies are needed because the diagnostic accuracy of PET or PET/CT with different tracers is not a measure of clinical effectiveness, and a good diagnostic performance does not necessarily result in improved outcomes in patients with brain tumors. Furthermore, several factors beyond the diagnostic performance may influence the choice of a specific PET tracer for evaluating brain tumors, such as tracer availability, radiation dose, and the presence of a cyclotron facility, as well as safety, legal, organizational, and economic aspects.

Several limitations of the selected meta-analyses should be discussed because they could hamper the ability to obtain definitive conclusions on the diagnostic performance or prognostic value of PET or PET/CT with different tracers in patients with brain tumors. In several selected meta-analyses, a limited number of original articles was included, and many of them had a small sample size. The limited number of cumulative patients available for the quantitative analysis may reduce the statistical power of the meta-analysis [33,34]. Furthermore, in some meta-analyses, a significant statistical heterogeneity among included studies was found, leading to a significant bias. This heterogeneity is likely due to differences in patient characteristics, methodological aspects (including different PET interpretation criteria), and study quality among the different included studies [33,34]. Some studies included in the meta-analyses used an imperfect reference standard (other than histology), which may have produced biased results. Moreover, publication bias (due to the higher probability of publishing studies reporting significant findings than those reporting non-significant results) has been detected in some meta-analyses.

Some indications of PET or PET/CT with different tracers in patients with brain tumors were not evaluated by published meta-analyses. In particular, there are no published meta-analyses about PET imaging with different tracers in evaluating other tumors of the central nervous system such as acustic neuromas, chordomas, craniopharyngiomas, ependymomas, optic nerve gliomas, medulloblastomas, meningiomas, pituitary adenomas, etc.

To date, there is an increased use of PET/CT with radiolabelled somatostatin analogues in patients with meningiomas due to the overexpression of somatostatin receptors by these tumors. Compared with standard MRI, somatostatin receptor PET may add valuable additional diagnostic information in these patients, particularly in the differential diagnosis of newly diagnosed brain lesions suspicious for meningiomas, delineation of meningioma extent for resection or radiotherapy planning, and the differentiation of tumor progression from a post-treatment change [35].

Furthermore, there are no meta-analyses about the diagnostic performance of PET with other emerging tracers for evaluation of brain tumors, such as radiolabelled prostate-specific membrane antigen (PSMA) [36] or fluorine-18 fluciclovine (^18^F-FACBC) [37], or with extensively studied tracers such as α-[^11^C]methyl-*L*-tryptophan [38].

Hybrid PET/MRI tomographs are now available for clinical use, and the role of PET/MRI using different PET tracers in brain tumors is promising, but more evidence-based data are needed in this setting [20].

Large multicenter prospective studies and, in particular, more cost-effectiveness analyses comparing different PET tracers and different neuroimaging modalities in patients with brain tumors are warranted. To this end, some cost-effectiveness analyses on the use of amino acid PET or PET/CT in brain tumors are already available, demonstrating that the combination of amino acid PET and MRI may be cost-effective for target selection in patients with suspicious glioma, for the surgical plan in patients with high-grade glioma, and for the evaluation of patients with suspicious recurrent high-grade glioma or brain metastasis [39,40,41,42].

## Figures and Tables

**Figure 1 ijms-20-04669-f001:**
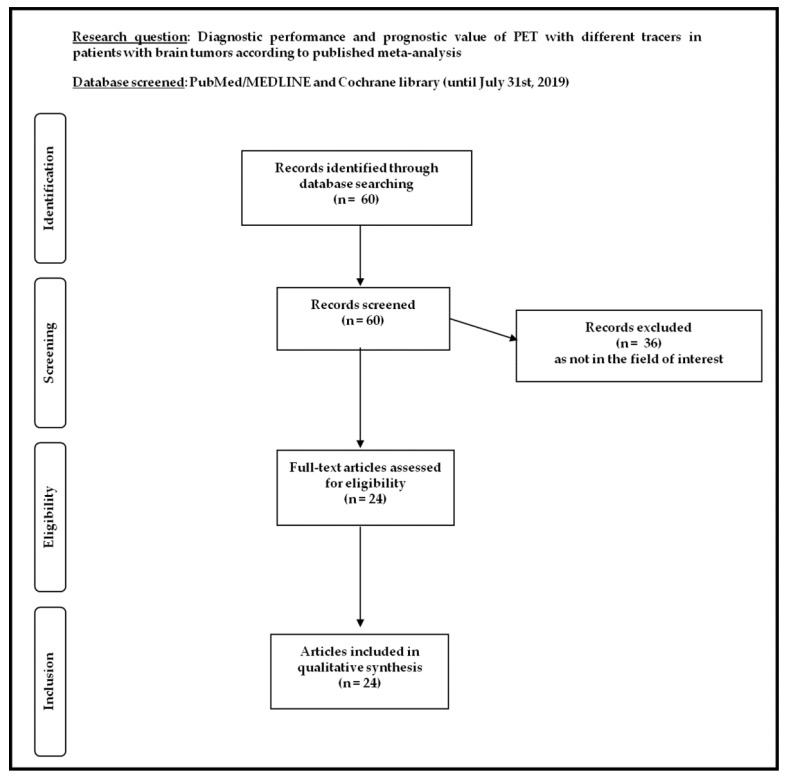
Flow chart of the search for meta-analyses on the diagnostic performance and prognostic value of positron emission tomography (PET) with different tracers in patients with brain tumors.

**Figure 2 ijms-20-04669-f002:**
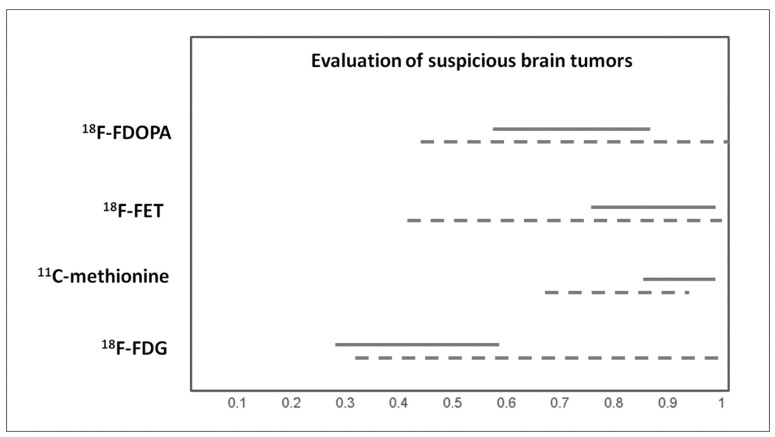
Graph showing 95% confidence intervals of sensitivity (continuous line) and specificity (dashed line) of PET with different tracers in evaluating suspicious brain tumors, according to published meta-analyses.

**Figure 3 ijms-20-04669-f003:**
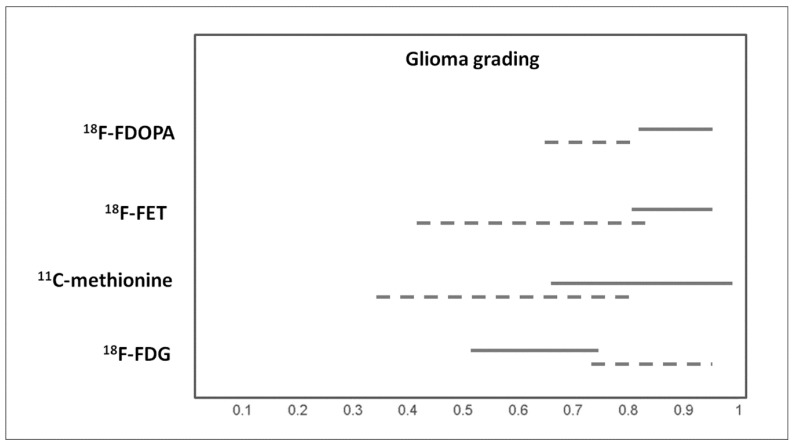
Graph showing 95% confidence intervals of sensitivity (continuous line) and specificity (dashed line) of PET with different tracers for glioma grading, according to published meta-analyses.

**Figure 4 ijms-20-04669-f004:**
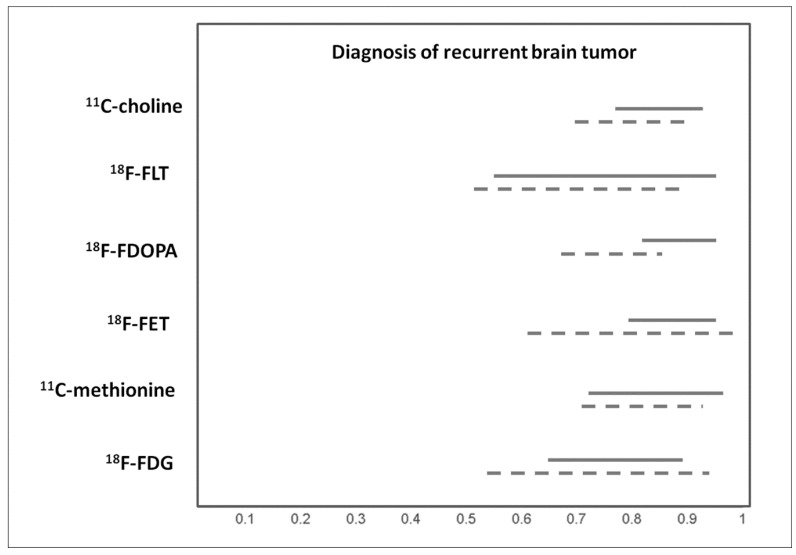
Graph showing 95% confidence intervals of sensitivity (continuous line) and specificity (dashed line) of PET with different tracers for diagnosis of recurrent brain tumor, according to published meta-analyses.

**Figure 5 ijms-20-04669-f005:**
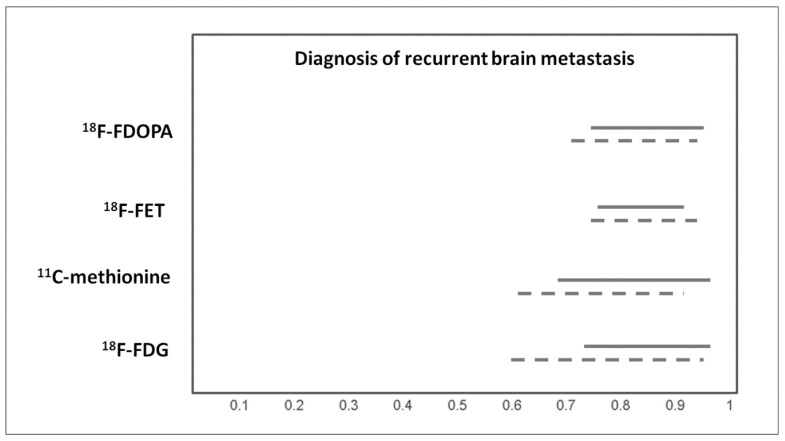
Graph showing 95% confidence intervals of sensitivity (continuous line) and specificity (dashed line) of PET with different tracers for diagnosis of recurrent brain metastasis, according to published meta-analyses.

**Table 1 ijms-20-04669-t001:** Characteristics and main findings of included meta-analyses on the diagnostic performance of PET or PET/computed tomography (CT) with different tracers in patients with brain tumors.

Indication	Tracer	Authors	Year	Articles Included	Patients Included	Sensitivity (95% CI)	Specificity (95% CI)	LR + (95% CI)	LR − (95% CI)	DOR (95% CI)
**Evaluation of Suspicious Primary Brain Tumor**	^18^F-FDG	Zhao et al. [28]	2014	3	127	43% (28–59)	74% (49–90)	1.7 (0.6–4.8)	0.77 (0.48–1.24)	NR
		Dunet et al. [25]	2016	5	119	38% (27–50)	86% (31–99)	2.7 (0.3–27.8)	0.72 (0.47–1.11)	4 (0–58)
	^11^C-methionine	Zhao et al. [28]	2014	2	85	95% (85–98)	83% (65–93)	5.5 (2.5–12.2)	0.07 (0.02–0.2)	NR
	^18^F-FET	Dunet et al. [31]	2012	5	224	82% (74–88)	76% (44–92)	3.4 (1.2–9.5)	0.24 (0.14–0.39)	14 (3–60)
		Dunet et al. [25]	2016	5	119	94% (79–98)	88% (37–99)	8.1 (0.8–80.6)	0.07 (0.02–0.30)	113 (4–2975)
	^18^F-FDOPA	Xiao et al. [8]	2019	5	46	71% (54–85)	86% (42–100)	3.7 (0.9–15.8)	0.36 (0.19–0.68)	10.88 (1.57–75.31)
**Glioma Grading**	^18^F-FDG	Dunet et al. [25]	2016	2	63	60% (mean TBR ≥1.4) 72% (max TBR ≥1.8)	91% (mean TBR ≥1.4) 73% (max TBR ≥1.8)	NR	NR	NR
		Katsanos et al. [11]	2019	13	680	63% (51–74)	89% (73–95)	5.2 (2.1–13)	0.42 (0.29–0.6)	12.4 (3.86–39.8)
	^11^C-methionine	Falk Delgado et al. [19]	2018	13	241	80% (66–88)	72% (62–81)	NR	NR	NR
		Katsanos et al. [11]	2019	8	191	94% (79–98)	55% (32–77)	2.1 (1.25–3.5)	0.11 (0.03–0.37)	18.25 (4.73–70.5)
	^18^F-FET	Dunet et al. [25]	2016	2	63	88% (mean TBR ≥2) 80% (max TBR ≥3)	73% (mean TBR ≥2) 82% (max TBR ≥3)	NR	NR	NR
		Katsanos et al. [11]	2019	7	259	88% (82–93)	57% (40–73)	2.1 (1.4–3.15)	0.2 (0.11–0.37)	10.16 (3.9–26.5)
	^18^F-FDOPA	Xiao et al. [8]	2019	7	219	88% (81–93)	73% (64–81)	2.9 (2.2–3.85)	0.16 (0.08–0.36)	25.87 (10.53–63.54)
**Glioma Delineation**	^11^C-methionine	Verburg et al. [21]	2017	5	NR	[HGG] 93.7%	[HGG] 61.3%	NR	NR	[HGG] 26.6
**Diagnosis of Recurrent Brain Tumor**	^18^F-FDG	Nihashi et al. [30]	2013	16	NR	77% (66–85)	78% (54–91)	3.4 (1.6–7.5)	0.3 (0.21–0.43)	NR
		Zhao et al. [28]	2014	20	643	75% (67–81)	79% (66–88)	3.5 (2.2–5.7)	0.32 (0.25–0.41)	NR
		Li et al. [26]	2015	22	NR	78% (69–85)	77% (66–85)	3.3 (2.2–5)	0.29 (0.20–0.42)	12 (6–22)
		Wang et al. [27]	2015	12	418	70% (64–75)	88% (80–93)	4 (2.1–7.5)	0.38 (0.29–0.51)	NR
		Furuse et al. [12]	2019	9	327	81% (67–90)	72% (64–79)	NR	NR	NR
	^11^C-methionine	Nihashi et al. [30]	2013	7	NR	[HGG] 70% (50–84)	[HGG] 93% (44–100)	[HGG] 10.3 (0.8–139.4)	[HGG] 0.32 (0.18–0.57)	NR
		Deng et al. [29]	2013	11	244	87% (81–91.8)	81.3% (71.5–88.8)	4.35 (2.8–6.8)	0.19 (0.13–0.29)	21.86 (10.7–44.5)
		Zhao et al. [28]	2014	8	238	92% (83–97)	87% (75–93)	6.8 (3.4–13.7)	0.09 (0.04–0.21)	NR
		Wang et al. [27]	2015	6	156	85% (76–91)	83% (71–92)	4.4 (2.5–7.7)	0.22 (0.13–0.35)	NR
		Xu et al. [18]	2017	29	899	88% (85–91)	85% (80–89)	5.3 (3.3–8.7)	0.16 (0.11–0.23)	35.3 (22.9–54.4)
		Furuse et al. [12]	2019	8	333	81% (73–87)	81% (74–87)	NR	NR	NR
	^18^F-FET	Yu et al. [16]	2018	27	NR	82% (79–84)	80% (76–83)	3.9 (3.0–5.1)	0.21 (0.17–0.27)	23.03 (14.42–36.77)
		Furuse et al. [12]	2019	3	138	91% (79–97)	95% (61–99)	NR	NR	NR
	^18^F-FDOPA	Yu et al. [16]	2018	21	NR	85% (81–88)	77% (74–81)	3.4 (2.8–4.3)	0.21 (0.16–0.29)	21.7 (12.61–37.33)
		Xiao et al. [8]	2019	13	318	92% (88–95)	76% (66–85)	2.9 (2–4.1)	0.13 (0.07–0.23)	29.65 (13–09–67–15)
	AA *	Kim et al. [9]	2019	6	212	89% (82–94)	88% (76–94)	7.3 (3.6–14.7)	0.12 (0.07–0.21)	60 (23–152)
	^18^F-FLT	Li et al. [26]	2015	5	NR	82% (51–95)	76% (50–91)	3.5 (1.6–7.7)	0.24 (0.08–0.70)	15 (4–56)
	^11^C-choline	Gao et al. [14]	2018	6	118	87% (78–93)	82% (69–91)	4.9 (2.6–9.1)	0.16 (0.09–0.29)	35.5 (11.7–107.7)
**Diagnosis of Brain Metastases**	^18^F-FDG	Li et al. [24]	2017	5	941	21% (13–32)	100% (99–100)	184.7 (24.8–1374)	0.79 (0.70–0.89)	235 (31–1799)
**Diagnosis of Recurrent Brain Metastases**	^18^F-FDG	Li et al. [17]	2018	6	NR	85% (77–94)	90% (83–96)	NR	NR	NR
		Suh et al. [13]	2018	5	NR	83% (74–92)	88% (81–95)	NR	NR	NR
		Furuse et al. [12]	2019	3	NR	91% (73–97)	80% (60–91)	NR	NR	NR
	^11^C-methionine	Li et al. [17]	2018	2	NR	86% (74–97)	79% (66–93)	NR	NR	NR
		Furuse et al. [12]	2019	4	NR	79% (67–87)	76% (61–87)	NR	NR	NR
	^18^F-FET	Li et al. [17]	2018	5	NR	83% (76–91)	89% (83–95)	NR	NR	NR
		Yu et al. [16]	2018	4	NR	80% (76–84)	79% (75–83)	3.9	0.24	19
	^18^F-FDOPA	Li et al. [17]	2018	2	NR	86% (74–97)	88% (79–97)	NR	NR	NR
		Yu et al. [16]	2018	2	NR	78% (73–82)	75% (71–89)	3	0.31	11
	AA *	Suh et al. [13]	2018	7	NR	84% (79–90)	85% (80–91)	NR	NR	NR
**Diagnosis of PCNSL**	^18^F-FDG	Zhou et al. [22]	2017	8	129	88% (80–94)	86% (73–94)	4 (2.3–6.9)	0.11 (0.04–0.32)	33.4 (10.4–107.3)
		Yang et al. [23]	2017	6	108	NR	NR	NR	NR	NR

Legend: LR+ = positive likelihood ratio; LR− = negative likelihood ratio; DOR = diagnostic odds ratio; 95% CI = 95% confidence interval; AA * = radiolabelled amino acid PET including radiolabelled methionine, fluoroethyltyrosine and fluorodihydroxyphenylalanine; NR = not reported; HGG = high grade gliomas only; PCNSL = primary central nervous system lymphoma; mean TBR = mean tumor-to-background uptake ratio; max TBR = maximum tumor-to-background uptake ratio.

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
