# Peer review of "Diagnostic Performance and Prognostic Value of PET/CT with Different Tracers for Brain Tumors: A Systematic Review of Published Meta-Analyses"

_ijms, 2019, doi:10.3390/ijms20194669_

Round 1
Reviewer 1 Report
The review paper summarizes published meta-analysis of PET tracers in diagnostic performance and prognostic evaluation of brain tumors. Several common-used PET tracers are included in the current review in terms of sensitivity, specificity in primary brain tumors, glioma grading, diagnosis of recurrent brain tumors, diagnosis of brain metastases, and recurrent brain metastases. The review provides supportive evidence-based data for better utilization of various PET tracers in the imaging of brain tumors. The paper is recommended for publication in the International Journal of Molecular Sciences after the authors address the following issues:
In addition to various PET tracers, different imaging modalities, such as PET alone compared to PET/CT or PET/MRI will affect the sensitivity and specificity. The authors should discuss these if applicable; The authors may discuss another amino acid PET tracer α-[11C]methyl-L-tryptophan, although this tracer may not be evaluated in a meta-analysis, it has extensively been studied in adult and pediatric brain tumors; Only limited brain tumor types are discussed in the review paper, not sure whether there are not any meta-analysis using these tracers in different brain tumors.Author Response
Reviewer's comment: In addition to various PET tracers, different imaging modalities, such as PET alone compared to PET/CT or PET/MRI will affect the sensitivity and specificity. The authors should discuss these if applicable.
Response: we agree with the Reviewer's comment. We have added a statement in the discussion underlining that in addition to various PET tracers, different imaging modalities, such as PET alone compared to PET/CT or PET/MRI will affect the sensitivity and specificity.
Reviewer's comment: The authors may discuss another amino acid PET tracer α-[11C]methyl-L-tryptophan, although this tracer may not be evaluated in a meta-analysis, it has extensively been studied in adult and pediatric brain tumors.
Response: We have added a statement in the discussion about α-[11C]methyl-L-tryptophan as PET tracer for brain tumors.
Reviewer's comment: Only limited brain tumor types are discussed in the review paper, not sure whether there are not any meta-analysis using these tracers in different brain tumors.
Response: we confirm that there are not further published meta-analyses about these tracers for other brain tumor types. We have added in the revised manuscript that "In particular, there are not published meta-analyses about PET imaging with different tracers in evaluating other tumors of the central nervous system as acustic neuromas, chordomas, craniopharyngiomas, ependymomas, optic nerve gliomas, medulloblastomas, meningiomas, pituitary adenomas, etc."
Reviewer 2 Report
It is an overall thorough review to the previous studies on PET imaging of brain tumors, including primary brain tumor, brain metastases, and tumor recurrence. The manuscript has been well organized and well written. Herein, the reviewer would have the following concerns:
1) For the better visualization and citation by peers, the reviewer would ask the authors to see if the sensitivity and specificity data of all reviewed tracers can also be summarized using comparison graphs, including mean values and statistical ranges.
2) Please check the correct or more accurate wording in English for "a not adequate", and "a non optimal" in Section 3.1 of the manuscript.
Author Response
Reviewer's comment: For the better visualization and citation by peers, the reviewer would ask the authors to see if the sensitivity and specificity data of all reviewed tracers can also be summarized using comparison graphs, including mean values and statistical ranges.
Response: we have added graph of sensitivity and specificity in the revised manuscript according to the reviewer's suggestions.
Reviewer's comment: Please check the correct or more accurate wording in English for "a not adequate", and "a non optimal" in Section 3.1 of the manuscript.
Response: we have corrected these statements in Section 3.1.